# ADAPTIVE PATH-INTEGRAL APPROACH FOR REPRESENTATION LEARNING AND PLANNING

**Jung-Su Ha, Young-Jin Park, Hyeok-Joo Chae, Soon-Seo Park & Han-Lim Choi**
Department of Aeropsace Engeneering, KAIST
Daejeon, Republic of Korea
`{{jsha, yjpark, hjchae, sspark}@lics., hanlimc@}kaist.ac.kr`

## ABSTRACT

We present a novel framework for representation learning that builds a low-dimensional latent dynamical model from high-dimensional *sequential* raw data, e.g., video. The framework builds upon recent advances in amortized inference methods that use a differentiable network to output samples from a variational distribution given observations as inputs, and takes advantage of the duality between control and inference to approximately solve the intractable inference problem using the path integral control approach. We also present an efficient planning method that exploits the learned low-dimensional latent dynamics.

## 1 APPROXIMATE INFERENCE VIA STOCHASTIC OPTIMAL CONTROL

In an approximate inference, it is known that a tighter evidence lower bound (ELBO) can be achieved by using multiple samples, $\mathbf{z}^{1:L}$, independently sampled from the proposal distribution $q(\mathbf{z})$:

$$\log p(\mathbf{x}) \geq \mathcal{L}^L \equiv \mathbb{E}_{\mathbf{z}^{1:L} \sim q(\cdot)} \left[ \log \frac{1}{L} \sum_{l=1}^{L} \frac{p(\mathbf{x}, \mathbf{z}^l)}{q(\mathbf{z}^l)} \right] \geq \mathcal{L}^{L-1}. \tag{1}$$

It is proven that the ELBO gets tighter as $L$ increases (Burda et al., 2016; Cremer et al., 2017). This class of multi-sample objectives, $\mathcal{L}^L$, is referred as a Monte Carlo objectives (MCOs) in the sense that it utilizes independent samples to estimate the marginal likelihood (Mnih & Rezende, 2016),

$$\hat{I}(\mathbf{z}^{1:L}) = \frac{1}{L} \sum_{l=1}^{L} \frac{p(\mathbf{x}, \mathbf{z}^l)}{q(\mathbf{z}^l)} = \frac{1}{L} \sum_{l=1}^{L} \frac{p(\mathbf{x}|\mathbf{z}^l)p(\mathbf{z}^l)}{q(\mathbf{z}^l)}, \ \mathbf{z}^l \sim q(\cdot), \ \forall l \in \{1, ..., L\}. \tag{2}$$

The performance of the MCO-based learning algorithm crucially depends on the variance of $\hat{I}(\mathbf{z}^{1:L})$, which can be reduced by decreasing the gap between the proposal distribution, $q(\mathbf{z})$, and the true posterior distribution, $p(\mathbf{z}|\mathbf{x})$; when $q(\mathbf{z}) = p(\mathbf{z}|\mathbf{x})$, the variance reduces to 0.

In this work, we consider latent variable models that have the continuous-time stochastic dynamics for latent variables, $\mathbf{z} \in \mathbb{R}^{d_z}$, i.e., the prior $p(\mathbf{z}_{[0,T]})$ is a probability measure of a following system:

$$d\mathbf{z}(t) = \mathbf{f}(\mathbf{z}(t))dt + \sigma(\mathbf{z}(t))d\mathbf{w}(t), \ \mathbf{z}(0) \sim p_0(\cdot), \tag{3}$$

where $\mathbf{w}(t)$ is a $d_u$-dimensional Wiener process, and a sequential observation $\mathbf{x}_{1:K}$ with $p(\mathbf{x}_{1:K}|\mathbf{z}_{[0,T]}) = \prod_{k=1}^{K} p(\mathbf{x}_k|\mathbf{z}(t_k))$, where $\{t_k\}$ is a sequence of discrete time points with $t_1 = 0, t_K = T$. For an approximate inference, the variational proposal distributions are parameterized as the distributions of a controlled continuous-time stochastic dynamical system with the controls, $\mathbf{u} \in \mathbb{R}^{d_u}$, and parameters of an initial state distribution, $q_0$, serving as the variational parameters, i.e., the proposal $q_{\mathbf{u}}(\mathbf{z}_{[0,T]})$ is a probability measure of a following system:

$$d\mathbf{z}(t) = \mathbf{f}(\mathbf{z}(t))dt + \sigma(\mathbf{z}(t))(\mathbf{u}(t)dt + d\mathbf{w}(t)), \ \mathbf{z}(0) \sim q_0(\cdot), \tag{4}$$

By applying Girsanov's theorem in Appendix A which provides the likelihood ratio between $p(\mathbf{z}_{[0,T]})$ and $q_{\mathbf{u}}(\mathbf{z}_{[0,T]})$, the ELBO is written as:

$$\mathcal{L} = \mathbb{E}_{q_{\mathbf{u}}(\mathbf{z}_{[0,T]})} \left[ \log p(\mathbf{x}_{1:K}|\mathbf{z}_{[0,T]}) + \log \frac{dp(\mathbf{z}_{[0,T]})}{dq_{\mathbf{u}}(\mathbf{z}_{[0,T]})} \right]$$

$$= \mathbb{E}_{q_{\mathbf{u}}(\mathbf{z}_{[0,T]})} \left[ \log p(\mathbf{x}_{1:K}|\mathbf{z}_{[0,T]}) + \log \frac{p_0(\mathbf{z}(0))}{q_0(\mathbf{z}(0))} - \frac{1}{2} \int_0^T ||\mathbf{u}(t)||^2 dt - \int_0^T \mathbf{u}(t)^T d\mathbf{w}(t) \right]. \quad (5)$$

Then, the problem of finding the optimal variational parameters $\mathbf{u}^*$ and $q_0$ (or equivalently, the best approximate posterior) can be formulated as a stochastic optimal control (SOC) problem:

$$\mathbf{u}^*, q_0^* = \operatorname*{argmin}_{\mathbf{u}, q_0} \mathbb{E}_{q_{\mathbf{u}}(\mathbf{z}_{[0,T]})} \left[ V(\mathbf{z}_{[0,T]}) + \frac{1}{2} \int_0^T ||\mathbf{u}(t)||^2 dt + \int_0^T \mathbf{u}(t)^T d\mathbf{w}(t) \right], \quad \textbf{(SOC)}$$

where $V(\mathbf{z}_{[0,T]}) \equiv -\log \frac{p_0(\mathbf{z}(0))}{q_0(\mathbf{z}(0))} - \sum_{k=1}^K \log p(\mathbf{x}_k|\mathbf{z}(t_k))$ serves as a state cost function of the SOC problem. Such equivalence between finding the optimal control and the posterior inference has been well known in literatures, e.g., Fleming & Mitter (1982); Todorov (2008); Kappen & Ruiz (2016). This work particularly adopts the adaptive path integral control method to obtain valid $q_{\mathbf{u}}$, which consequently results in a valid proposal $q$ for MCO estimation.

## 2 PROPOSED METHOD

### 2.1 ADAPTIVE PATH INTEGRAL AUTOENCODER

The above (**SOC**) is a class of SOC problems of which the objective function can be written as a KL divergence form by Girsanov's theorem:

$$J = KL\left(q_{\mathbf{u}}(\mathbf{z}_{[0,T]})||p^*(\mathbf{z}_{[0,T]})\right) - \log\xi, \quad (6)$$

where $p^*$, represented as $dp^*(\mathbf{z}_{[0,T]}) = \exp(-V(\mathbf{z}_{[0,T]}))dp(\mathbf{z}_{[0,T]})/\xi$, is the probability measure induced by the optimally-controlled trajectories and $\xi \equiv \int \exp(-V(\mathbf{z}_{[0,T]}))dp(\mathbf{z}_{[0,T]})$ is a normalization constant (see Appendix A for details). By applying Girsanov's theorem again, the optimal trajectory distribution is expressed as:

$$dp^*(\mathbf{z}_{[0,T]}) \propto dq_{\mathbf{u}}(\mathbf{z}_{[0,T]}) \exp\left(-S_{\mathbf{u}}(\mathbf{z}_{[0,T]})\right), \quad (7)$$

$$S_{\mathbf{u}}(\mathbf{z}_{[0,T]}) = V(\mathbf{z}_{[0,T]}) + \frac{1}{2} \int_0^T ||\mathbf{u}(t)||^2 dt + \int_0^T \mathbf{u}(t)^T d\mathbf{w}(t). \quad (8)$$

This yields that the optimal trajectory distribution can be approximated by sampling a set of trajectories according to the controlled dynamics with $\mathbf{u}(t)$, i.e. $\mathbf{z}_{[0,T]}^l \sim q_{\mathbf{u}}(\cdot)$, and assigning their importance weights as $\tilde{w}^l = \frac{\exp(-S_{\mathbf{u}}(\mathbf{z}_{[0,T]}^l))}{\sum_{i=1}^L \exp(-S_{\mathbf{u}}(\mathbf{z}_{[0,T]}^i))}$, $\forall l \in \{1, ..., L\}$. Similar to the variance of (2), the variance of importance weights decreases as the control input $\mathbf{u}(\cdot)$ gets closer to the true optimal control input $\mathbf{u}^*(\cdot)$ and it reduces to 0 when $\mathbf{u}(t) = \mathbf{u}^*(t, \mathbf{z}(t))$ (Thijssen & Kappen, 2015).

Path-Integral control is a sampling-based SOC method, which approximates the optimal trajectory distribution with weighted sample trajectories using (7)–(8) and updates control parameters based on moment matching of $q_{\mathbf{u}}$ to $p^*$. Let $\mathbf{u}_{ff}(t)$ and $\mathbf{K}(t)$ represent feedforward control and feedback gain, respectively. We use a standardized linear feedback controller, where the control input has a form as:

$$\mathbf{u}(t) = \mathbf{u}_{ff}(t) + \mathbf{K}(t)\Sigma^{-1/2}(t)(\mathbf{z}(t) - \mu(t)), \quad (9)$$

where $\mu(t) = \sum_{l=1}^L \tilde{w}^l \mathbf{z}^l(t)$ and $\Sigma(t) = \sum_{l=1}^L \tilde{w}^l (\mathbf{z}^l(t) - \mu(t))(\mathbf{z}^l(t) - \mu(t))^T$ are the mean and covariance of the state w.r.t. the approximated optimal trajectory distribution, respectively. Suppose we have a set of trajectories and their weights obtained by the control policy with $\mathbf{u}(t) = \bar{\mathbf{u}}_{ff}(t) + \bar{\mathbf{K}}(t)\bar{\Sigma}^{-1/2}(t)(\mathbf{z}(t) - \bar{\mu}(t))$. Then, the path integral control theorem in the Appendix B gives the control policy update rule with the adaptation rate $\eta$ as:

$$\mathbf{u}_{ff}(t)dt = \bar{\mathbf{u}}_{ff}(t)dt + \bar{\mathbf{K}}(t)\bar{\Sigma}^{-1/2}(t)(\mu(t) - \bar{\mu}(t))dt + \eta \sum_{l=1}^L \tilde{w}^l d\mathbf{w}^l(t), \quad (10)$$

$$\mathbf{K}(t)dt = \bar{\mathbf{K}}(t)\bar{\Sigma}^{-1/2}(t)\Sigma^{1/2}(t)dt + \eta \sum_{l=1}^L \tilde{w}^l d\mathbf{w}^l(t) \left(\Sigma^{-1/2}(t)(\mathbf{z}^l(t) - \mu(t))\right)^T, \quad (11)$$

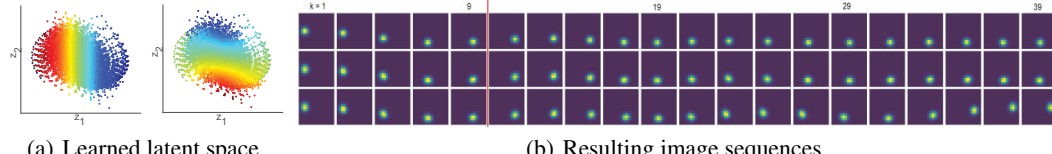

(a) Learned latent space    (b) Resulting image sequences

Figure 1: (a) The inferred latent states colored by angles (left) and angular velocities (right) of the ground truth. (Warm colors represent higher values.) (b) Top: ground truth, Middle: reconstruction ($k \leq 10$) & prediction ($k > 10$), Bottom: reconstruction ($k \leq 10$) & planning ($k > 10$).

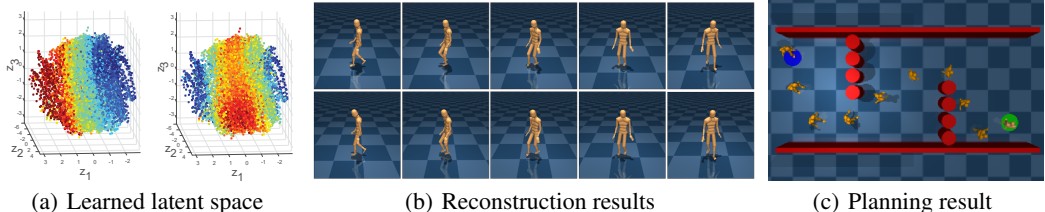

(a) Learned latent space    (b) Reconstruction results    (c) Planning result

Figure 2: (a) The inferred latent states colored by yaw rates (left) and forward velocities (right) of the ground truth. (Warm colors represent higher values.) (b) Top: ground truth, Bottom: reconstruction. (c) The starting point and goal region are blue and green circles, respectively.

In addition, the initial state distribution also can be updated into $q_0(\cdot) = \mathcal{N}(\cdot; \hat{\mu}_0, \hat{\Sigma}_0)$:

$$\hat{\mu}_0 = \sum_{l=1}^{L} \tilde{w}^l \mathbf{z}^l(0), \ \hat{\Sigma}_0 = \sum_{l=1}^{L} \tilde{w}^l (\mathbf{z}^l(0) - \hat{\mu}_0)(\mathbf{z}^l(0) - \hat{\mu}_0)^T. \tag{12}$$

Starting from the passive dynamics (3), i.e. $\mathbf{u} = 0$ and $q_0 = p_0$, the variational parameters are gradually updated in order for the resulting trajectory distribution to be close to the posterior distribution. After $R$ adaptations, the MCO and its gradient are then estimated by:

$$\hat{\mathcal{L}}^L = \log \frac{1}{L} \sum_{l=1}^{L} \exp(-S_{\mathbf{u}}(\mathbf{z}_{[0,T]}^l)), \ \nabla_\theta \hat{\mathcal{L}}^L = -\sum_{l=1}^{L} \tilde{w}^l \nabla_\theta S_{\mathbf{u}}(\mathbf{z}_{[0,T]}^l), \tag{13}$$

where $\theta$ denotes the parameter of the latent variable model, i.e. $\mathbf{f}, \sigma, p_0(\mathbf{z})$ and $p(\mathbf{x}|\mathbf{z})$. Because all procedures in path integral adaptation and MCO construction are differentiable, they can be implemented by a fully differentiable network with $R$ recurrences, which we named Adaptive Path Integral Autoencoder (APIAE). We implemented APIAE with Tensorflow (Abadi et al., 2015); the pseudo code of APIAE is shown in the Appendix C and the implementation code is available at *https://github.com/yjparkLiCS/APIAE*.

## 2.2 PLANNING WITH LEARNED MODEL

The learned latent dynamical model can be exploited to generate a high-dimensional motion plan with the forward operation of APIAE. Suppose that, for the planning problem, the objective function, $C_k(\mathbf{x}_k)$, encodes given task specifications, e.g., a desired/undesired configuration, collision with obstacles, etc. Then, the state cost function of the path integral adaptation is changed into:

$$V(\mathbf{z}_{[0,T]}) = \mathbb{E}_{p(\mathbf{x}_{1:K}|\mathbf{z}_{[0,T]})} \left[ \sum_{k=1}^{K} C_k(\mathbf{x}_k) \right]. \tag{14}$$

After the adaptations with (14), the resulting plan can be sampled from the generative model as $\mathbf{x}_{1:K} \sim p(\cdot|\mu_{[0,T]})$. Note that the time interval, $t_k - t_{k-1}$, or the trajectory length, $K$, used in model learning and planning can differ because we are dealing with continuous-time dynamics.

## 3 EXPERIMENTS

The first experiment addresses the system identification and planning of pendulum dynamics with sequences of $16 \times 16$ images. Fig. 1(a) shows the constructed 2-D latent space; each point represents

the posterior mean of the observation data. Interestingly, each dimension approximately encoded the negative angle and the negative angular velocity, respectively. The middle row of Fig. 1(b) shows images predicted by our generative model, and they are almost same to the ground truth. We formulated planning problems in the image space of which objectives were to put the pendulum to the right-most position. It is shown from Fig. 1(b) that the resulting motion plans achieved the desired configuration successfully in a dynamically natural way.

The second experiment addresses a motion planning of a humanoid robot with 62-D configuration space. We utilized human's walk and turn data from the Carnegie Mellon University motion capture database and the DeepMind Control Suite (Tassa et al., 2018) for parsing and visualization. A 3-D latent space was considered in this example and, as shown in Fig. 2(a), the high-D data were well embedded into the low-dimensional latent space. Fig. 2(b) shows that APIAE successfully reconstructed the data. We then formulated planning problems, where the cost function penalized collision with any obstacle, large yaw rate, and distance from the goal. Fig. 2(c) shows that the proposed method successfully generated the natural and collision-free motion toward the goal.

We would refer the readers to the Appendix D and the supplementary video at *https://youtu.be/NM6dxxfh37U* for more experimental details and results with clearer visualization.

## 4 DISCUSSION

There have been various studies on the representation learning of dynamical systems by optimizing the VAE's ELBO, e.g., (Krishnan et al., 2017; Johnson et al., 2016; Karl et al., 2017; Fraccaro et al., 2017), and more recently, the MCO with resampling, e.g., (Maddison et al., 2017; Le et al., 2018; Naesseth et al., 2017). We would like to emphasize that the proposed method is a complementary technique to these existing methods; the adaptation procedure of APIAE can play a role in constructing more expressive/accurate posterior distribution approximations from the proposal, just as IWAE (Burda et al., 2016) constructs better approximations of posterior distributions from the VAE's proposals with multiple samples Cremer et al. (2017). The experiments showed that initializing the proposal with the passive trajectory distribution leads to sufficiently accurate posterior distribution approximations within a few improvement steps in the considered problems, but the variational proposal distribution also can be introduced to address more challenging problems.

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

## A  OBJECTIVE FUNCTION OF STOCHASTIC OPTIMAL CONTROL PROBLEM

Suppose that an objective function of a stochastic optimal control (SOC) problem is given as:

$$J = \mathbb{E}_{q_{\mathbf{u}}} \left[ \int_0^T V(\mathbf{z}(t)) + \frac{1}{2} ||\mathbf{u}(t)||^2 dt \right], \tag{15}$$

where $q_{\mathbf{u}}$ is the probability measures induced by the controlled trajectories, $\mathbf{z}_{[0,T]} \equiv \{\mathbf{z}(t); \ \forall t \in [0,T]\}$. The first and second terms in the integral encodes a state cost and penalizes control input effort, respectively. The objective of the SOC problem is to find the optimal control sequence $\mathbf{u}^*(t)$ as well as the initial state distribution $q_0$, with which the trajectory distribution of (4) minimizes the objective function (15).

The following theorem implies that the control penalty term in (15) can be interpreted as the KL divergence between distributions of controlled and uncontrolled trajectories.

**Theorem 1 (Girsanov's Theorem (modified from Gardiner et al. (1985)))** *Suppose $p$ and $q_{\mathbf{u}}$ are the probability measures induced by the trajectories of* (3) *and* (4)*, respectively. Then, the Radon-Nikodym derivative of $q_{\mathbf{u}}$ with respect to $p$ is given by*

$$\frac{dp(\mathbf{z}_{[0,T]})}{dq_{\mathbf{u}}(\mathbf{z}_{[0,T]})} = \frac{p_0(\mathbf{z}(0))}{q_0(\mathbf{z}(0))} \exp\left( -\frac{1}{2} \int_0^T ||\mathbf{u}(t)||^2 dt - \int_0^T \mathbf{u}(t)^T d\mathbf{w}(t) \right), \tag{16}$$

*where $\mathbf{w}(t)$ is a Wiener process for simulating $q_{\mathbf{u}}$.*

With Girsanov's theorem, the objective function (15) is rewritten in the form of KL divergence:

$$\begin{aligned}
J &= \mathbb{E}_{q_{\mathbf{u}}} \left[ \int_0^T V(\mathbf{z}(t)) + \frac{1}{2} ||\mathbf{u}(t)||^2 dt \right] \\
&= \mathbb{E}_{q_{\mathbf{u}}} \left[ \int_0^T V(\mathbf{z}(t)) dt + \log \frac{dq_{\mathbf{u}}(\mathbf{z}_{[0,T]})}{dp(\mathbf{z}_{[0,T]})} - \log \frac{q_0(\mathbf{z}(0))}{p_0(\mathbf{z}(0))} \right] \\
&= \mathbb{E}_{q_{\mathbf{u}}} \left[ \log \frac{dq_{\mathbf{u}}(\mathbf{z}_{[0,T]})}{dp(\mathbf{z}_{[0,T]}) \exp(-V(\mathbf{z}_{[0,T]}))/\xi} - \log \xi \right] \\
&= KL\left( q_{\mathbf{u}}(\mathbf{z}_{[0,T]}) || p^*(\mathbf{z}_{[0,T]}) \right) - \log \xi, \tag{17}
\end{aligned}$$

where $V(\mathbf{z}_{[0,T]}) \equiv \int_0^T V(\mathbf{z}(t)) dt - \log \frac{q_0(\mathbf{z}(0))}{p_0(\mathbf{z}(0))}$ is a trajectory state cost and $\xi \equiv \int \exp(-V(\mathbf{z}_{[0,T]})) dp(\mathbf{z}_{[0,T]})$ is a normalization constant. Note that the second term in the exponent of (16) disappears when taking the expectation w.r.t. $q_{\mathbf{u}}$, i.e. $\mathbb{E}_{q_{\mathbf{u}}}[\int_0^T \mathbf{u}(t)^T d\mathbf{w}(t)] = 0$, because $\mathbf{w}(t)$ is a Wiener process for simulating $q_{\mathbf{u}}$.

## B  DERIVATION OF PATH INTEGRAL ADAPTATION

From the trajectories sampled with $q_{\mathbf{u}}(\cdot)$, the path integral control provides how to compute the optimal control $\mathbf{u}^*(t)$ based on the following theorem.

**Theorem 2 (Main Theorem (Thijssen & Kappen, 2015))** *Let $f : \mathbb{R} \times \mathbb{R}^{d_z} \to \mathbb{R}$, and consider the process $f(t) = f(t, \mathbf{z}(t))$ with $\mathbf{z}_{[0,T]} \sim q_{\mathbf{u}}(\cdot)$. Then,*

$$\langle (\mathbf{u}^* - \mathbf{u})f \rangle (t) = \lim_{\tau \to t} \left\langle \frac{\int_t^\tau f(s) d\mathbf{w}(s)}{\tau - t} \right\rangle, \tag{18}$$

*where $\langle Y(t) \rangle \equiv \mathbb{E}_{q_{\mathbf{u}}}[\tilde{w}_{\mathbf{u}} Y(t)]$, $\tilde{w}_{\mathbf{u}} = \frac{\exp(-S_{\mathbf{u}}(\mathbf{z}_{[0,T]}))}{\mathbb{E}_{q_{\mathbf{u}}}[\exp(-S_{\mathbf{u}}(\mathbf{z}_{[0,T]}))]}$ for any process $Y(t)$.*

Suppose the current control policy is parameterized with $n_b$ basis functions $\bar{h}(t, \mathbf{z}) : \mathbb{R} \times \mathbb{R}^{d_z} \to \mathbb{R}^{n_b}$ as:

$$\bar{\mathbf{u}}(t, \mathbf{z}(t)) = \bar{\mathbf{A}}(t) \bar{h}(t, \mathbf{z}(t)), \tag{19}$$

where $\bar{\mathbf{A}}(t) : \mathbb{R} \to \mathbb{R}^{d_u \times n_b}$ is the control policy parameter and let the optimal parameterized control policy be $\mathbf{u}^* = \mathbf{A}^*(t)h(t, \mathbf{z}(t))$. Then, Theorem 2 can be rewritten as:

$$\mathbf{A}^*(t) \langle h \otimes h \rangle (t) = \bar{\mathbf{A}}(t) \langle \bar{h} \otimes h \rangle (t) + \lim_{\tau \to t} \left\langle \frac{\int_t^\tau d\mathbf{w}(s) \otimes h(s)}{\tau - t} \right\rangle. \tag{20}$$

Because we can utilize only a finite number of samples to approximate the optimal trajectory distribution, it is more reasonable to update the control policy parameter with some small adaptation rate, than to estimate it at once. Similar to Ruiz & Kappen (2017), we use a standardized linear feedback controller w.r.t. the target distribution, i.e.,

$$h(t, \mathbf{z}(t)) \equiv \left[ 1; \Sigma^{-1/2}(t)(\mathbf{z}(t) - \mu(t)) \right], \tag{21}$$

where $\mu(t) = \langle \mathbf{z}(t) \rangle$ and $\Sigma(t) = \langle (\mathbf{z}(t) - \mu(t))(\mathbf{z}(t) - \mu(t))^T \rangle$ are the mean and covariance of the state w.r.t. the optimal trajectory distribution estimated at the previous iteration. Then, the control input has a form as:

$$\mathbf{u}(t) = \mathbf{u}_{ff}(t) + \mathbf{K}(t)\Sigma^{-1/2}(t)(\mathbf{z}(t) - \mu(t)), \tag{22}$$

where the parameter, $\mathbf{A}(t) = [\mathbf{u}_{ff}(t), \mathbf{K}(t)]$, represents feedforward control signal and feedback gain.

Suppose we have a set of trajectories and their weights obtained by the parameterized policy, $\bar{\mathbf{u}}(t) = \bar{\mathbf{A}}(t)\bar{h}(t, \mathbf{z}(t))$. Then, based on (20), the control policy parameters can be updated as follows:

$$\mathbf{u}_{ff}(t)dt = \bar{\mathbf{u}}_{ff}(t)dt + \bar{\mathbf{K}}(t)\bar{\Sigma}^{-1/2}(t)(\mu(t) - \bar{\mu}(t))dt + \eta \langle d\mathbf{w}(t) \rangle, \tag{23}$$

$$\mathbf{K}(t)dt = \bar{\mathbf{K}}(t)\bar{\Sigma}^{-1/2}(t)\Sigma^{1/2}(t)dt + \eta \left\langle d\mathbf{w}(t) \left( \Sigma^{-1/2}(t)(\mathbf{z}(t) - \mu(t)) \right)^T \right\rangle, \tag{24}$$

where $\eta$ is an adaptation rate[1]. Note that the adaptation of two terms can be done independently, because $\langle h \otimes h \rangle (t) = I$. Beside the control policy adaptation, the initial state distribution, $p_0$, can be updated as well:

$$\hat{\mu}_0 = \langle \mathbf{z}(0) \rangle, \ \hat{\Sigma}_0 = \langle (\mathbf{z}(0) - \hat{\mu}_0)(\mathbf{z}(0) - \hat{\mu}_0)^T \rangle, \tag{25}$$

where the updated trajectory distribution starts from $q_0(\cdot) = \mathcal{N}(\cdot; \hat{\mu}_0, \hat{\Sigma}_0)$.

## C  ALGORITHMIC DETAILS

The pseudo code of APIAE training is shown in Algorithm 1. The training procedure of APIAE consists of three steps: inference, reconstruction, and model update. Given the observation data, the algorithm approximates the posterior distribution using path integral adaptation method in the inference step (line 3–8), reconstructs data from the inferred latent trajectory in the reconstruction step (line 9–12), and updates the model parameter according to the MCO gradients in the model update step (line 13). In detail, each function in the algorithm works as follows.

- SIMULATE(): First, it independently samples $L$ initial latent states, $\mathbf{z}_1^{1:L} \sim \mathcal{N}(\hat{\mu}_0, \hat{\Sigma}_0)$. Using the current control policy $\mathbf{A}_{[0,T]}$ and Wiener processes, $\{\mathbf{w}_{[0,T]}\}^{1:L}$, it simulates the dynamics (4) from the initial states, $\mathbf{z}_1^{1:L}$. The stochastic simulation can be done as:

$$\mathbf{z}(t + \delta t) = \mathbf{z}(t) + \mathbf{f}(\mathbf{z}(t))\delta t + \sigma(\mathbf{z}(t))(\mathbf{u}(t)\delta t + W_t \sqrt{\delta t}),$$

  where $\delta t$ is a predefined time interval and $W_t \sim \mathcal{N}(0, I_{d_u})$.
- COST(): Each trajectory is evaluated according to (8).
- IMPROVE(): Improved control policy and initial state distribution are computed using (10)-(11) and (12).

---

[1]At the first iteration, $\bar{\mathbf{u}}_{ff}(t) = 0, \bar{\mathbf{K}}(t) = 0, \bar{\mu}(t) = 0, \bar{\Sigma}(t) = I$ and $q_0 = p_0$.

---

**Algorithm 1** Training of Adaptive Path Integral Autoencoder

---

**Input**: Dataset of observation sequences, $\mathcal{D} = \{\mathbf{x}_{1:K}^{(i)}\}_{i=1,...,N}$.
 Latent and observation models, $\mathbf{f}(\mathbf{z}), \sigma(\mathbf{z}), p_0(\mathbf{z})$ and $p(\mathbf{x}|\mathbf{z})$, parameterized by $\theta$.

1: **while** $notConverged()$ **do**
2:  Sample datapoint $\mathbf{x}_{1:K}$ from $\mathcal{D}$
3:  Initialize $\hat{\mu}_0 \leftarrow \mu_0, \hat{\Sigma}_0 \leftarrow \Sigma_0, \mathbf{A}_{[0,T]} \leftarrow 0$
4:  **for** $r \in \{1,...,R\}$ **do**
5:   $\{\mathbf{z}_{[0,T]}, \mathbf{w}_{[0,T]}\}^{1:L} \leftarrow \text{SIMULATE}(\hat{\mu}_0, \hat{\Sigma}_0, \mathbf{A}_{[0,T]})$
6:   $S_{\mathbf{u}}^{1:L} \leftarrow \text{COST}(\mathbf{x}_{1:K}, \{\mathbf{z}_{[0,T]}, \mathbf{w}_{[0,T]}\}^{1:L}, \mathbf{A}_{[0,T]})$
7:   $\hat{\mu}_0, \hat{\Sigma}_0, \mathbf{A}_{[0,T]} \leftarrow \text{IMPROVE}(\{S_{\mathbf{u}}, \mathbf{z}(0), \mathbf{w}_{[0,T]}\}^{1:L}, \mathbf{A}_{[0,T]})$
8:  **end for**
9:  $\{\mathbf{z}_{[0,T]}, \mathbf{w}_{[0,T]}\}^{1:L} \leftarrow \text{SIMULATE}(\hat{\mu}_0, \hat{\Sigma}_0, \mathbf{A}_{[0,T]})$
10:  $S_{\mathbf{u}}^{1:L} \leftarrow \text{COST}(\mathbf{x}_{1:K}, \{\mathbf{z}_{[0,T]}, \mathbf{w}_{[0,T]}\}^{1:L}, \mathbf{A}_{[0,T]})$
11:  $\hat{w}^{1:L} \leftarrow \exp(-S_{\mathbf{u}}^{1:L})/\sum_l \exp(-S_{\mathbf{u}}^l)$
12:  $\hat{\mathcal{L}} = \log \frac{1}{L} \sum_l \exp(-S_{\mathbf{u}}^l), \nabla_\theta \hat{\mathcal{L}} \leftarrow -\sum_l \hat{w}^l \nabla_\theta S_{\mathbf{u}}^l$
13:  Update $\theta$ with $\nabla_\theta \hat{\mathcal{L}}$ using SGD          ▷ gradients are aggregated across mini-batches.
14: **end while**

---

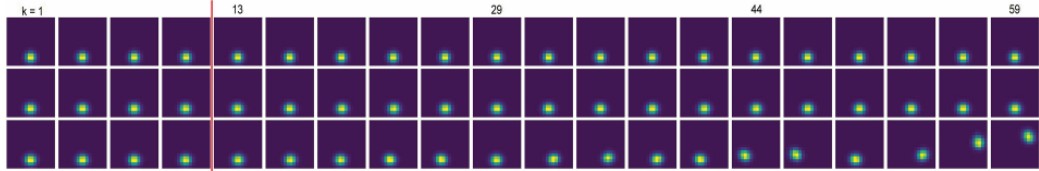

Figure 3: Resulting image sequences for the swing-up task. Top: ground truth, Middle: reconstruction ($k \leq 10$) & prediction ($k > 10$), Bottom: reconstruction ($k \leq 10$) & plaining ($k > 10$).

The path integral adaptation and the MCO construction steps of APIAE can be seen as encoding and decoding procedures of autoencoders, respectively, motivating the name "adaptive path integral autoencoder."

The inference, reconstruction, and gradient backpropagation of APIAE can operate independently for each of the $L$ samples. Consequently, the computational cost grows linearly with the number of samples, $L$, and the number of adaptations, $R$. As implemented in IWAE Burda et al. (2016), we replicated each observation data $L$ times and the whole operations were parallelized with GPU. Note that APIAE is not the first algorithm that implements the optimal planning/control algorithm into a fully-differentiable network; similar structures have been proposed by Tamar et al. (2016); Okada et al. (2017), but their application domains are different to this work.

# D  EXPERIMENTAL DETAILS AND MORE RESULTS

## D.1  PENDULUM

The pendulum dynamics is represented by the second order differential equation for angle of the pendulum, $\psi$, as:

$$\ddot{\psi}(t) = -a\sin(\psi) - b\dot{\psi}, \tag{26}$$

where the parameters $a$ and $b$ were set to 9.8 and 1, respectively. We simulated the pendulum dynamics from random initial states with the time interval, $\delta t = 0.1$, and trajectory length, $K = 10$. Sequences of images corresponding to the pendulum state was made as $16 \times 16$ sized image, and those videos were the training data of APIAE, i.e., $\mathbf{x}_k$ lied in 256-D observation space.

A 2-D latent state space was used, and the dynamics was parameterized as:

$$\mathbf{f}(\mathbf{z}(t)) = \begin{bmatrix} z_2(t) \\ f(\mathbf{z}(t)) \end{bmatrix}, \sigma(\mathbf{z}(t)) = \begin{bmatrix} 0 \\ 1 \end{bmatrix}, \tag{27}$$

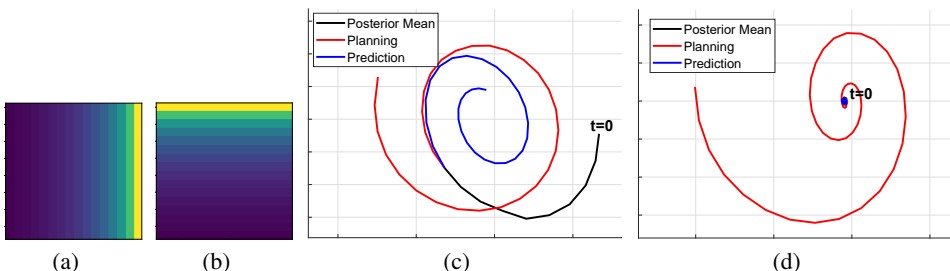

Figure 4: Objective functions for (a) the swing-right task and (b) the swing-up task. (c) & (d) The mean of posterior, planned and predicted latent trajectories for each task.

where $f(\cdot)$ is a neural network having 128 hidden units with ReLU activation. For the stochastic simulation, we simply chose $t_k = (k-1)\delta t$, and $\delta t = T/(K-1)$. For the observation model, we considered a neural network with Bernoulli outputs as:

$$\log p(\mathbf{x}|\mathbf{z}) = \sum_{i=1}^{256} x_i \log y_i + (1 - x_i) \log(1 - y_i), \tag{28}$$

where $\mathbf{y} = g(z_1)$ and $g(\cdot)$ is also a neural network having 128 hidden units with ReLU activation and a 256-D output layer with sigmoid activation. We found that initializing dynamics network to be stable results in the more stable learning, therefore the dynamics network was initialized by the supervised learning with transition data from stable linear system. That is, the initial dynamics operates similarly to a stable linear system. Finally, we trained APIAE with $L = 50$ and $R = 3$.

Fig. 3 shows the another planning result for the swing-up task. The objectives of the planning problems were to minimize the cross entropy between the goal image, shown in Fig. 4(a)-(b), and the last frame of the sequence ($K = 39$ & $59$, respectively); the desired motion plans were to put the pendulum to the right and upright position, respectively. For the planning, we used $R = 30$ and $L = 10000$; the smaller number of adaptations $R$ and samples $L$ also can generate the plan successfully, but we found larger $R$ and $L$ show smoother resulting trajectories. Note that the planning procedure is operated with single batch, which allows to use such large $L$ and $R$. By observing 10 frames before planning, the initial latent state for the planning is localized. It is shown from Fig. 3 that the resulting motion plans achieved desired configuration successfully, while maintaining dynamic feasibility. Figs. 4(c)-(d) also show that the learned model was able to infer the latent state given the observation, and able to predict and plan the future state.

### D.2 HUMANOID MOTION RECONSTRUCTION AND PLANNING

The 62-D configurations of the Mocap data consist of angles of all joints, roll and pitch angles, vertical position of the root, yaw rate of the root, and horizontal velocity of the root. The global (horizontal) position and heading orientation are not encoded in the generative model (only velocities are encoded), but they can be recovered by integration if the trajectory is given. The original data were written at 120 Hz, and we down-sampled the data to 20 Hz and cut the data every 10 time steps, i.e. $\delta t = 0.05$, $K = 10$.

A 3-D latent state space was used in this example, and the dynamics was parameterized as a neural network, $\mathbf{f}(\cdot) \in \mathbb{R}^3$, having 2 hidden layers where each layer has 128 hidden units with ReLU activation and a 3-D output layer without activation, and $\sigma(\mathbf{z}(t)) = \begin{bmatrix} 0 & 0 \\ 1 & 0 \\ 0 & 1 \end{bmatrix}$. For the stochastic simulation, we simply chose $t_k = (k-1)\delta t$, and $\delta t = T/(K-1)$. For the observation model, we considered a neural network with Gaussian outputs as:

$$p(\mathbf{x}|\mathbf{z}) = \mathcal{N}(\mathbf{x}; g(\mathbf{z}), I_{62}), \tag{29}$$

where $g(\cdot)$ is a neural network having 2 hidden layers where each layer has 128 hidden units with ReLU activation and a 62-D output layer without activation. As in the pendulum experiment, we

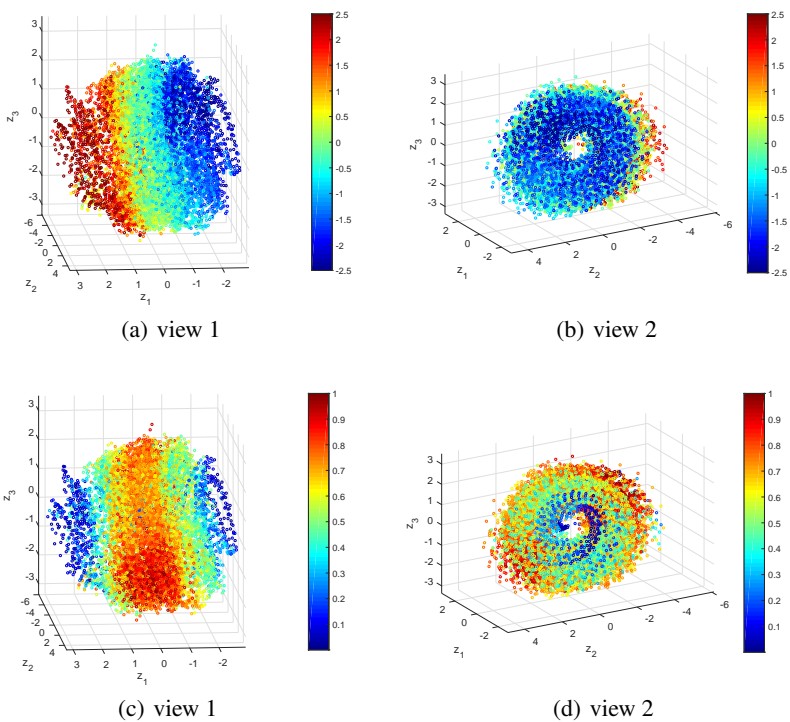

(a) view 1          (b) view 2

(c) view 1          (d) view 2

Figure 5: The learned latent space colored by (a)-(b) yaw rate and (c)-(d) forward velocity of the ground truth.

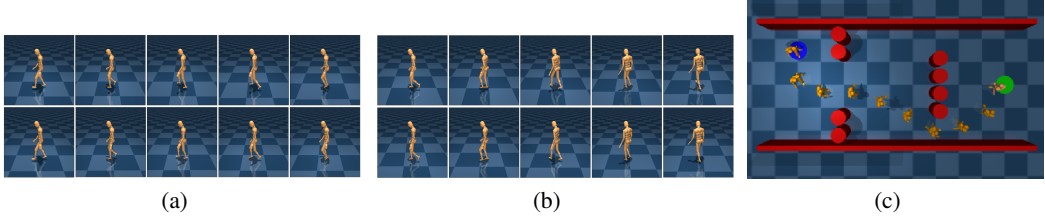

(a)          (b)          (c)

Figure 6: (a)-(b) Reconstruction results. (b) Planning result.

initialized dynamic network such that it operates like a stable linear system for the stable learning. Finally, we set $L = 300$, $R = 3$ and $K = 10$ for the training, and $L = 10000$, $R = 30$, $K = 400, 300$ for the planning, respectively. Fig. 5 illustrates the posterior mean states of the training data. The motion data were arranged in the order of the yaw rates along the major axis of the ellipse (Fig. 5(a)-(b)). Also, the motions with lower forward velocities were embedded into smaller radius cycles (Fig. 5(c)-(d)). Because there was a negative correlation between forward velocity and the magnitude of yaw rate in the training data, they formed a rugby ball shaped manifold in the latent space. Finally, Fig. 6 depicts additional reconstruction and planning results.

