# OpenReview forum: "Adaptive Path-Integral Approach for Representation Learning and Planning"
_ICLR.cc/2018/Workshop — Accept_

### Official Review · AnonReviewer2 · 2018-03-09
**Seems interesting, but probably inscrutable to non-specialists**

**Rating:** 6
**Confidence:** 3

**Review:**

Pros:
+ Integrates priors for sequence-modeling into a VAE-like method
+ Reduces the ELBO maximization to an optimal control problem and uses sophisticated path integral control methods to solve this
+ Seems to learn some meaningful representation in a toy problem

Cons:
+ Paper is poorly written and only accessible to an extremely specialized community
+ I have doubts as to the necessity of the very fancy techniques deployed here compared to regular VAEs

As someone interested in both deep generative models and path-integral methods, I found this work pretty interesting.  However, I found the paper very difficult to read and struggled to understand the basic concept promoted here.  After much frustration, I would summarize the key ideas in the following steps:

1) The high-level idea is to use a VAE that somehow exploits the structure of modeling continuous trajectories
2) For this reason, the variational proposal distributions are parameterized as the distributions of a controlled continuous-time stochastic dynamical system with the controls serving as the variational parameters—i.e., the proposals are distributions over paths resulting from rolling out a noisy dynamical system with some controls
3) The latent prior in the VAE (p(z)) is chosen to be the distribution of the same stochastic dynamical system, but with zero controls
4) Stochastic calculus is used to equate maximization of the ELBO with the solution of a stochastic optimal control problem in a “canonical” form (i.e., with a quadratic control penalty)
5) Stochastic calculus is again used to obtain a concrete form for the importance weights (i.e., likelihood ratio, or Radon-Nikodym derivative) used to evaluate the ELBO for a given control sequence
6) Some path integral control method is used to optimize the ELBO in the “canonical optimal control” form

The presentation sorely lacks a concise summary of the type I have just given. Presented in the current form, the work just sounds like a jumble of equations and terms from variational inference, stochastic calculus, and optimal control, without any guiding motivation. To list some other annoyances regarding the presentation:

+ Point (3) above is very unclear, but I assume it is implied by “measure, p, induced by (3) with u(t) = 0”.  Just saying “p” is ambiguous, and mentioning this important point in a parenthetical expression is also unwise.
+ The end of section 1 should be at the beginning.  It makes little sense to mention the purpose of doing some complicated optimal control manipulations only after doing them.  This is also phrased in a very odd, ambiguous way “Note that if we relate… becomes equivalent to the posterior inference.”  Why not just clearly state, “We reduce posterior inference to doing optimal control… etc.,” instead of leaving room for ambiguity?

That said, I do appreciate the work for the fact that it does show how to blend stochastic optimal control and variational inference in a nice mathematical framework.  My only real technical comment is that I am a bit skeptical of approximating the partition function by a few samples, as is done here to compute the importance weights, though I guess this is probably common in path integral control.

My other doubt is whether the complexity of this approach is really justified by the problem.  There are no baselines in the experiments, and it seems like a regular VAE (with no latent structure) would probably be able to learn similar latent representations with similar reproduction quality.  I would even go as far as to argue that VAEs themselves are overkill with respect to simple Gaussian mixture models in many cases.  So, practically speaking, I don’t foresee the community at large rushing to implement this for their own applications.

In summary, in order to have a larger community impact, I would recommend focusing more on the concrete benefits of this approach for trajectory modeling vs. naive methods, and I would refocus the presentation with an emphasis on the overall flow of the method.

---

### Official Review · AnonReviewer3 · 2018-03-12
**Interesting but difficult to read paper combining stochastic optimal control and variational inference**

**Rating:** 6
**Confidence:** 3

**Review:**

## Paper summary and general comments

This paper presents an interesting fusion of ideas from stochastic optimal control and parametric variational inference to suggest a method for learning low-dimensional dynamic latent variable models for sequential data. The topic is relevant to ICLR and the specific application of the path integral control framework within a parametric variational inference setting seems to be novel, although as discussed below it would be helpful for the specific contribution being made by the authors to be more clearly stated. The experiments although with relatively simple models help to back up the claims about the proposed method and seem of adequate complexity / depth for a short workshop submission.

Although the ideas in the paper seem interesting and relevant, as currently written it is difficult to follow. Some of this is due to minor grammatical mistakes and typos (see detailed list below) though these should be simple to fix. More critically the description of the background material is currently quite hard to parse. Though the appendices help to clarify some details, the main paper should still ideally be understandable on its own, with the appendices used to add further detail and expand on aspects only given in overview. More detailed specific comments and areas needing clarification are given below.

It would also be of benefit for the paper to more clearly state the specific contribution being made and describe its relation to previous work. Given the space constraints understandably it is infeasible to include an extensive literature review / discussion of related work, but a succint statement of the specific contribution being made and how this builds on a few key references would significantly improve the paper. In particular
it would help to explain the relation of the proposal to the following papers *in the main paper (not appendix)*

  [1] Particle smoothing for hidden diffusion processes: Adaptive path integral smoother'. Ruiz and Kappen (2017)
  [2] Adaptive Importance Sampling for Control and Inference. Kappen and Ruiz (2016)
  [3] Value iteration networks. Tamat et al. (2016)
  [4] Path integral network - end-to-end differentiable optimal control. Okada et al. (2017)

Of these the [1], [3] and [4] are all briefly referenced in the appendices but not in the main paper, and [2] is not referred to at all but seems to be closely related to the proposed method.

## Specific comments and questions

### Abstract

In the second sentence the clause 'The framework builds upon recent advances in the amortized inference
that constructs a fully-differentiable network' - it is not clear how the relative clause following 'that' relates to the preceding part of sentence. My understanding (possibly incorrect) is that you were trying express something along the lines of 'recent advances in amortized inference methods that use a differentiable network to output the parameters a variational distribution given latent state values as inputs' more succinctly, but as it stands it is difficult to parse what you have written.

### Page 1

In the fourth paragraph you refer to equation (3) before its introduced in the text which is a bit confusing. Ideally rearrange so the SDE model is introduced before being referenced or rephrase the section beginning 'is induced by (3)'.

Some intuition for equation (4) would be helpful and how it relates to the earlier discussion of variational inference in latent variable models.

It would help to have more clarification of where the different terms in equations (5) and (6) arise from in the main text particularly the path integral term integrating the control function against the Wiener process. Referring derivations in the appendices is fine but it would help to give an overview of the role of each term.

### Page 2

Previous working relating optimal stochastic control to posterior inference e.g. Flemming & Mitter (1982), Kappen & Ruiz (2016) should be reference when this link is made in the text.

'With this new distribution, the compensated term, $\log{\frac{q_0(\mathbf{z}(0))}{p_0(\mathbf{z}_0)}$, needs to be added to (6)' - this needs some explanation, it is not immediately clear why this term is needed though it can be inferred with a bit of work. See also comment regarding Appendix 2 below.

### Page 3

The explanation and discussion of the pendulum dynamics planning problem experiment is very brief. You claim 'the resulting motion plan achieved the desired configuration successfully while maintaining dynamic feasibility' - it is not immediately clear what 'dynamic feasibility' in in this context. I would suggest moving these results and discussion to the appendix and presenting results only for state inference for the pendulum experiments in the main paper - the second motion planning experiment already provides an example of a planning application.

Some clarification of why it is considered that 'the high-D data were well embedded in the low-dimensional latent space' for the humanoid robot motion planning experiment would be beneficial - from Fig 2(a) I am inferring (possibly incorrectly) this is as the learnt manifold seems to be self-organised according to the (unknown at inference time) yaw rates and forward velocities however this could be made more explicit if it is correct. Further the organisation according to the forward velocity seems to be non-linear - some comment on this might be interesting e.g. is the colour coding actually of a (signed) velocity component here or the velocity magnitude (speed)?

### Appendix 1

Given the derivation in equation (15) specifically uses the Radon-Nikodym derivative of $q_{\mathbf{u}}$ with respect to $p$ rather than of $p$ with respect to $q_{\mathbf{u}$ it would be clearer to define the former in Theorem 1 / equation (14) rather than the latter.

The derivation here relies on the path integral of the control term against the Wiener process being zero in expectation - although this is stated in a footnote this is easy to miss as it comes at the end of the section - I only realised why it was valid to drop the term here after reading through the derivation in Thisjenn and Kappen (2015). I would move the footnote into the main text. Further the footnote as currently written does not make sense 'because $\mathbf{w}$ is for $q_{\mathbf{u}}$' needs rewriting.

### Appendix 2

Again the rationale for introducing the 'compensated term' ('compensation term' might be a better name) when changing the initialisation distribution is not very clearly explained here. It would help if it was explicitly stated that $S_{\mathbf{u}}$ is *defined* as being the negative logarithm of the Radon-Nikodym derivative of $p^\ast$ with respect to $q_{\mathbf{u}}$ from which the need to include the additional term when changing the initialisation distribution is more apparent. Although this is implied by the relationship in equation (5), it appears as if the definition of $S_{\mathbf{u}}$ is in equation (6) rather than this being a specific instatiation for a particular choice of $q_{\mathbf{u}}$.

### Appendix 3

Line numbers in Algorithm 1 are out of sync with description.

In description of the SAMPLE routine it is stated that $L$ independent Wiener process samples $\mathbf{w}_{[0,T]}$. As up to this point there is no discussion of discretisation of the Wiener process it is a bit unclear what is meant here; further in the subsequent definition of the SIMULATE routine, despite being defined as taking the Wiener process realisations as an argument, the actual description describes an Euler-Maruyama discretisation of the underlying SDE with the sampling of the Wiener process terms redefined in terms of draws from a standard normal here, making the previous definition of them being sampled in the SAMPLE routine redundant. One possibility would be to combine the SAMPLE and SIMULATE steps in to a single SIMULATE definition which includes the sampling of the initial state and Wiener process initialisations (and returns both the simulated state sequences $\mathbf{z}_{1:K}$ and Wiener process realisations $\mathbf{w}_{[0,T]}$).

It would also help to be clear here that the discretisation here introduces some approximation error (even if negligible in the context of the whole method, up to this point the discussion had been purely in terms of continuous stochastic processes, which potentially adds unnecessary complexity when in reality all simulations are performed in discrete time) and to be more explicit about the specific discretisation scheme used.

## Minor comments / typos

### Abstract

  * 'builds upon recent advances in the amortized inference' -> 'builds upon recent advances in amortized inference'
  * 'takes advantage of the duality between control and inference to solve the intractale inference problem' - 'solve' is a bit of a strong claim here - 'approximately solve' would be better
  * 'We also present the efficient planning method' -> 'We also present an efficient planning method'

### Page 1

  * 'this multi-sample objective $\mathcal{L}^L$, is referred as Monte Carlo objectives' -> 'this multi-sample objective $\mathcal{L}^L$, is referred to as a Monte Carlo objective'
  * 'Consider a continuous-time stochastic dynamics' -> 'Consider a continuous time stochastic dynamic' or 'Consider the continuous-time stochastic dynamics'
  * 'There is a class of stochastic optimal control problems of which objective function can be written' - 'There is a class of stochastic optimal control problems of which the objective function can be written'
  * 'by the Girsanov's theorem' -> 'by Girsanov's theorem' or 'by the Girsanov theorem'
  * 'see the Appendix A for details' -> 'see Appendix A for details'

### Page 2

  * 'which consequently results in valid proposal $q$ for MCO construction' -> 'which consequently results in a valid proposal $q$ for MCO construction'
  * 'are the mean and variance of the state' -> 'are the mean and covariance of the state'

### Page 3

  * In Figures 1(a) and 2(a) it would be better to use a perceptually uniform and sequential colormap, to allow the figure to still be interpreted if printed in grayscale and to improve comprehension for readers with color vision deficiencies.
  * Caption of Figure 1(b) 'plaining' probably meant to be 'planning' (?)
  * 'is complimentary technic to these exisiting methods' -> 'is a complimentary technique to these existing methods'
  * 'APIAE can play a role in constructing more expressive posterior distribution' -> 'APIAE can play a role in constructing more expressive posterior distributions' but actually 'posterior distribution approximations' may be closer to what is intended?
  * 'just as IWAE (Burda et al. 2016) constructs better distribution on VAE with multiple samples' - this needs rephrasing - what does 'better distribution' mean in this context? In IWAE multiple samples are used to construct an estimate of a tighter lower bound to the marginal likelihood compared to the the standard VAE evidence lower bound estimator, which can facilitate learning variational posterior approximations which better approximates the target posterior distribution.
  * 'The experiments showed that the passive trajectory distribution is an enough initial proposal to be improved in the considered problems' - this doesn't make sense - specifically 'is an enough initial proposal' is bad English and too vague. Perhaps you meant something along the lines of 'The experiments showed that initialising the proposal with the passive trajectory distribution lead to a sufficiently low-variance estimators to facilitate the initial learning steps in the considered problems'?

---

### Author Response · Authors · 2018-04-29
**Response to reviewers**

First of all, we really appreciate the Reviewers’ time and effort. We believe that the reviewers’ constructive comments have been very helpful to improve our paper. We have modified the manuscript with addressing the reviewers’ comments as thoroughly as possible; in particular, the flow of the paper has been changed, as the reviewers suggested, in order to increase its readability. There were some comments we didn't address yet, but our forthcoming full-paper will include them with more extensive quantitative and comparative analysis.

---

### Decision · Program_Chairs · 2018-03-20
**ICLR 2018 Workshop Acceptance Decision**

**Decision:**

Accept

**Comment:**

Congratulations, your paper was accepted to the ICLR workshop.